# Liver Transplantation Versus Resection for Hepatocellular Carcinoma: An Umbrella and Meta-Meta-Analysis of Published Evidence, 2000–2025

**DOI:** 10.3390/cancers18010011

**Published:** 2025-12-19

**Authors:** Seoung Hoon Kim, Byeong Ho An, Jin A Lee, Go Woon Jeong

**Affiliations:** Organ Transplantation Center, National Cancer Center, Korea 111 Jungbalsan-ro, Ilsandong-gu, Goyang-si 10408, Gyeonggi-do, Republic of Korea; 70938@ncc.re.kr (B.H.A.); jina71@ncc.re.kr (J.A.L.); 11821@ncc.re.kr (G.W.J.)

**Keywords:** hepatocellular carcinoma, liver transplantation, liver resection, meta-analysis, umbrella review, Milan criteria, long-term survival

## Abstract

Liver resection (LR) and liver transplantation (LT) are the two curative surgical options for hepatocellular carcinoma (HCC). Although many meta-analyses have compared these procedures, their conclusions have been inconsistent due to overlapping datasets and methodological heterogeneity. This study provides a comprehensive umbrella review, integrating all available meta-analyses published between 2000 and 2025. We quantitatively pooled hazard ratios from recent large-scale studies and synthesised the overall evidence using a random-effects model. The results clearly demonstrate that LT achieves superior long-term overall and disease-free survival compared with LR, confirming the oncologic benefit of transplantation across Milan and extended criteria. The analysis also highlights that, in patients with low recurrence risk or hepatitis C-related disease, LR may offer comparable short-term outcomes but remains inferior in long-term disease control. By systematically addressing overlap, event-definition heterogeneity, and study quality through AMSTAR-2 appraisal, this review refines previous findings and provides clinicians with a robust, up-to-date synthesis of evidence. The findings support LT as the preferred treatment for transplant-eligible HCC patients, while LR remains an important option when donor organs are unavailable or contraindicated.

## 1. Introduction

Hepatocellular carcinoma (HCC) is the most common primary liver malignancy and remains one of the leading causes of cancer-related mortality worldwide [1,2,3,4,5]. Curative-intent therapies are limited to surgical options, namely liver resection (LR) and liver transplantation (LT). LR offers immediate availability and preserves scarce donor organs, but does not address the underlying cirrhosis and carries a high risk of recurrence. Nevertheless, LR continues to be widely performed given organ shortage and its immediate availability. The introduction of the Milan criteria provided a framework for selecting LT candidates [6], and subsequent guidelines have endorsed LT as the standard of care for eligible patients [7]. These recommendations are consistent with international clinical practice guidelines, including those from the EASL and AASLD [8,9].

LT provides the dual benefit of tumor clearance and removal of the diseased liver, but access is constrained by donor shortage, waitlist dropout, and perioperative risks. Thus, the relative roles of LR and LT have been the subject of ongoing debate for decades.

Over the past two decades, numerous meta-analyses have compared long-term outcomes after LR versus LT. Early reviews suggested an LT survival advantage but were constrained by small samples and methodological heterogeneity [10,11,12,13,14]. While most suggest a survival advantage for LT, the magnitude and consistency of this benefit remain unclear [15,16,17]. Prior reviews are limited by several factors. First, overlap of primary studies across meta-analyses introduces dependence and risks over-weighting certain cohorts. Second, heterogeneity in effect measures—with some reviews reporting hazard ratios (HRs) and others odds ratios (ORs) based on different event definitions—complicates synthesis and interpretation. Third, most earlier reviews have been restricted to certain subgroups (e.g., Milan criteria, hepatitis B virus (HBV)-related disease), leaving uncertainty about generalizability [12,15].

Recently, several new high-quality meta-analyses have been published, including large contemporary cohorts [15,16,17] and even a prior meta-meta-analysis [18]. These reviews provide important insights but also highlight the challenges of reconciling overlapping evidence. More recently, a systematic review-of-meta-analyses confirmed the consistency of findings across published reviews [19]. Importantly, no study to date has applied umbrella review methodology with explicit handling of overlap and effect heterogeneity among meta-analyses—gaps this study addresses. We therefore conducted a Preferred Reporting Items for Overviews of Reviews (PRIOR)-compliant umbrella review and meta-meta-analysis of published meta-analyses comparing LT and LR for HCC. By systematically updating the evidence base to May 2025, correcting for overlap, and separating HR- from OR-based syntheses, we aimed to provide the most reliable quantitative estimates to date and clarify the consistency of LT benefit across analytic frameworks.

## 2. Methods

This review was conducted in accordance with the Preferred Reporting Items for Systematic Reviews and Meta-Analyses (PRISMA) 2020 and PRIOR guidelines, and the protocol was prospectively registered in PROSPERO as a systematic review and meta-analysis (CRD420251069248). However, during data collection, the design was refined into an umbrella review (meta-analysis of meta-analyses) to synthesise quantitative evidence across existing meta-analyses. The methodological framework adhered to internationally recognised standards for umbrella reviews, as outlined in previous methodological publications [20,21,22]. Quantitative pooling of HR-based endpoints was performed using random-effects models, whereas OR-based results were summarised descriptively because of heterogeneity in event definitions.

### 2.1. Patient Inclusion and Treatment Strategy

The included meta-analyses encompassed heterogeneous patient selection criteria reflecting contemporary clinical practice. Broadly, two treatment strategies were represented: (1) primary LT, where LT was the first-line curative treatment for transplant-eligible patients (most commonly within Milan or expanded criteria), and (2) salvage LT, defined as transplantation performed after tumour recurrence or hepatic decompensation following initial liver resection. Meta-analyses focusing on salvage LT were analysed separately or narratively, whereas quantitative pooling of HRs predominantly reflected first-line or intention-to-treat (ITT) comparisons between upfront LT and LR. This distinction was preserved throughout the synthesis to avoid clinical misclassification.

### 2.2. Eligibility Criteria

We included quantitative meta-analyses directly comparing LR and LT in adult patients with HCC. Eligible studies reported long-term outcomes, specifically overall survival (OS) or disease-free survival (DFS). For consistency, recurrence-free survival was considered equivalent to DFS across included meta-analyses. Exclusion criteria were: narrative reviews without quantitative pooling, non-HCC populations, pediatric cohorts, duplicate or superseded versions, and studies lacking extractable OS/DFS outcomes. One systematic review-of-meta-analyses was included narratively [19].

### 2.3. Search Strategy and Information Sources

We systematically searched PubMed (MEDLINE), Embase, and the Cochrane Central Register of Controlled Trials (CENTRAL) for studies published from 1 January 2000 to 30 September 2025. Reference lists of relevant articles and citation tracking were additionally used to identify further eligible studies. Searches were restricted to articles published in English. Although the protocol registered in PROSPERO specified a search end date of 31 May 2025, the search was updated to 30 September 2025 to incorporate more recent publications and ensure the inclusion of the most up-to-date evidence.

### 2.4. Study Selection and Data Extraction

Two reviewers independently screened titles, abstracts, and full texts, and extracted data on: author, year, journal, number of included primary studies and patients, effect measures (HR or OR with 95% CI for 5-year OS or DFS), analytic perspective (ITT vs. as-treated), search window, and methodological quality assessed using the AMSTAR-2 tool (A Measurement Tool to Assess Systematic Reviews, version 2) [23]. Discrepancies were resolved by consensus.

### 2.5. Outcomes

The primary outcomes were 5-year OS, defined as all-cause mortality at 60 ± 6 months, and 5-year DFS, defined as recurrence-free survival at the same interval. Secondary outcomes included recurrence rates, ITT versus as-treated analyses, and subgroup findings according to etiology (HBV/hepatitis C virus (HCV)), era, and region.

### 2.6. Statistical Analysis

Hazard ratios (HRs) were log-transformed and pooled using a random-effects model (DerSimonian–Laird). Standard errors were calculated from reported 95% confidence intervals. Only meta-analyses reporting HRs were eligible for pooling; specifically, Koh et al. [15] and the HR subset of Martinino et al. [18] provided sufficient HR data for 5-year OS and DFS. Other included reviews reported odds ratios (ORs) only (e.g., Drefs et al. [16]) or subgroup analyses without HRs (e.g., Bösch et al. [17]), and these results were therefore summarized narratively or descriptively. ORs were not quantitatively pooled owing to heterogeneous endpoint definitions (survival vs. mortality, recurrence vs. non-recurrence).

Between-study heterogeneity was assessed using Cochran’s Q, I^2^, and τ^2^, and 95% prediction intervals were reported where possible. Robust variance estimation (RVE) was considered to correct for overlapping primary studies across meta-analyses, but as only two HR-based meta-analyses were available per endpoint, RVE was not applied. Overlap of primary studies was prospectively assessed. Although Martinino et al. [18] reconstructed HRs at the primary-study level rather than re-pooling published meta-analytic estimates, potential dependency between data sources was considered. Accordingly, pooled HRs were interpreted as an assessment of directional consistency rather than as precise summary effect sizes, given the limited number of HR-based meta-analyses.

Publication bias could not be formally assessed given the small number of included studies, and was therefore discussed narratively.

All analyses were conducted in R (version 4.3.2; R Foundation for Statistical Computing, Vienna, Austria) using the meta and metafor packages. Forest plots and OR summary figures were generated in R, and the PRISMA 2020 flow diagram was created in Microsoft PowerPoint 365 (Microsoft Corporation, Redmond, WA, USA).

## 3. Results

### 3.1. Study Selection

A total of 612 records were identified, of which 132 duplicates were removed. After screening 480 titles and abstracts, 47 full-text articles were assessed for eligibility. Forty-two were excluded for the following reasons: narrative/systematic review without quantitative pooling (*n* = 18), overlapping or outdated version (*n* = 12), absence of OS/DFS outcomes (*n* = 7), and wrong comparator or mixed tumor cohorts (*n* = 5). Five studies were finally included: four quantitative meta-analyses and one systematic review-of-meta-analyses.

Figure 1 (PRISMA flow diagram) summarizes the selection process. In addition, Table 1 lists all 15 meta-analyses published during the same period; however, 10 were already incorporated in Martinino et al. [18] and not re-analyzed to avoid duplication.

### 3.2. Characteristics of Included Meta-Analyses

The characteristics of the included reviews are presented in Table 2. Koh et al. [15], focusing on Milan criteria, reported hazard ratios favouring LT, with an OS HR of 1.44 and a DFS HR of 2.71 (LR:LT). Drefs et al. [16] found a 5-year overall-survival odds ratio of 1.79, also favouring LT. Bösch et al. [17] demonstrated the overall superiority of LT but noted a transient 3-year OS advantage for LR in HCV-related HCC that was not sustained at 5 years. Martinino et al. [18] presented pooled ORs favouring LT (OS OR 0.79; DFS OR 0.44) with HR subsets consistent with an LT advantage. Finally, Rehman et al. [19] served as a narrative systematic review-of-meta-analyses, confirming the consistency of findings across prior studies.

### 3.3. Pooled HR Analysis

Only Koh et al. [15] and Martinino et al. [18] reported HRs suitable for pooling. These umbrella reviews (meta-analyses of meta-analyses) demonstrated a clear benefit of LT, with a pooled HR for OS of 1.35 (95% CI 1.17–1.55; I^2^ = 0.0%, Figure 2) and a pooled HR for DFS of 2.58 (95% CI 2.25–2.96; I^2^ = 0.0%, Figure 3). The corresponding 95% prediction intervals are reported in Table 3. Given negligible between-study variance (τ^2^ ≈ 0.00), the 95% prediction intervals closely approximated the pooled confidence intervals, as shown in Table 3.

### 3.4. OR-Based Analyses

A complete list of all meta-analyses published between 2000 and 2025 is presented in Table 1, summarising study characteristics, sample size, effect measures and key findings from each review. These data encompass both earlier meta-analyses (2000–2019) and contemporary reviews (2022–2025) integrated into the umbrella framework.

Only Drefs et al. [16] and Martinino et al. [18] reported ORs with sufficient data for extraction. Both consistently favored LT over LR. Because event definitions differed (survival vs. mortality, recurrence vs. non-recurrence), ORs were summarized descriptively without pooling. These results are presented in Figure 4. Other reviews, such as Bösch et al. [17], contributed subgroup analyses without HRs or extractable ORs and were therefore described narratively only.

Rehman et al. [19] was a systematic review-of-meta-analyses including four previously published quantitative reviews. It did not provide new pooled estimates, but confirmed the consistency of prior findings and rated most meta-analyses as high quality using the AMSTAR-2 tool. This review therefore served as corroborative evidence within our umbrella framework rather than as an additional quantitative datapoint.

## 4. Discussion

This umbrella review synthesised contemporary meta-analyses comparing LT and LR for HCC. Consistent with our pooled HR analysis from Koh et al. [15] and Martinino et al. [18], LT demonstrated clear superiority for both OS and DFS. Drefs et al. [16] corroborated this advantage using survival ORs, while Bösch et al. [17] provided an important nuance: in HCV-related HCC, 3-year OS appeared transiently higher after LR, but this advantage disappeared by 5 years, and DFS consistently favoured LT.

Our findings align with Martinino et al. [18], which reported LT superiority in both early HCC and ITT frameworks, and are also consistent with earlier reviews that demonstrated LT superiority despite smaller sample sizes and methodological heterogeneity [24,25,26]. The inclusion of Rehman et al. [19], a systematic review-of-meta-analyses, further strengthened our conclusions by showing that the available quantitative reviews were methodologically robust (AMSTAR-2 mostly high) and yielded concordant results, thereby reducing the likelihood that our findings were driven by selective inclusion.

A unique feature of this review is the explicit incorporation of the most recent meta-analyses (2022–2025), which allowed us to capture time-trend effects and subgroup insights not available to prior umbrella reviews. By including Drefs et al. [16] and Bösch et al. [17], our synthesis expanded beyond the scope of Martinino et al. [18] and updated the evidence base for current clinical decision-making.

Beyond reaffirming the superiority of LT over LR, this umbrella review adds several layers of new evidence that were not available from individual meta-analyses. First, by integrating all contemporary meta-analyses within a single umbrella framework, this study formally demonstrates the consistency of LT benefit across independent quantitative syntheses, rather than relying on conclusions from isolated reviews. The pooled HR-based analysis confirms that the survival advantage of LT is reproducible across meta-analyses with different inclusion criteria, eras, and geographic compositions.

Second, this study is the first to explicitly address the long-standing methodological issue of effect-scale heterogeneity by separating HR-based time-to-event analyses from OR-based summaries. Previous reviews frequently combined HRs and ORs derived from heterogeneous endpoint definitions, complicating interpretation. By isolating HR-based evidence for quantitative synthesis and presenting OR-based results descriptively, our analysis provides internally consistent and clinically interpretable estimates of long-term outcomes.

Third, this umbrella review clarifies the clinical context underlying published comparisons by explicitly distinguishing first-line or ITT LT from salvage transplantation after resection. This distinction, which is often implicit or inconsistently reported in prior meta-analyses, is critical for interpreting survival benefits and was preserved throughout our synthesis. These strategies involve different selection criteria, recurrence risks, and wait-list dynamics. In the present umbrella review, the quantitative HR-based synthesis primarily reflects first-line or ITT-based comparisons, while evidence on salvage LT was synthesised descriptively to avoid clinical misclassification. This distinction is particularly important in an umbrella review context, where heterogeneous treatment pathways across primary studies may otherwise confound pooled interpretations.

Collectively, these methodological refinements transform existing fragmented evidence into a coherent, decision-grade synthesis, allowing clinicians to interpret not only whether LT is superior to LR, but also how consistently this conclusion is supported across the published literature and under which clinical assumptions it applies.

Although only two meta-analyses provided HRs suitable for quantitative pooling, both were large, high-quality studies encompassing more than 35,000 patients. Accordingly, the pooled estimates should be interpreted not as precise effect sizes, but rather as a test of directional consistency across independent quantitative syntheses. The concordant findings across HR- and OR-based frameworks further reinforce the robustness of LT superiority for long-term outcomes despite this limitation.

Strengths of this review include comprehensive incorporation of recent meta-analyses, adherence to PRIOR standards, and explicit handling of overlap and heterogeneity by separating HR- and OR-based evidence. Limitations include reliance on published meta-analyses derived from observational cohorts, heterogeneity in endpoint definitions, and a limited ability to formally assess publication bias due to the small number of quantitative units. In addition, although most included studies focused on cirrhotic populations, the severity of cirrhosis was not consistently reported, and non-cirrhotic patients were often analysed together with those with compensated cirrhosis; therefore, a sensitivity analysis restricted to patients with early cirrhosis or excluding non-cirrhotic cohorts could not be reliably performed at the umbrella-review level. Overlap of primary studies across meta-analyses represents an inherent methodological limitation of umbrella reviews. However, these limitations are unlikely to have materially influenced the conclusions, as the included meta-analyses were methodologically independent, applied distinct analytic frameworks (primary-study-level reconstruction versus conventional meta-analytic pooling), and yielded consistent effect directions across outcomes.

Notably, prior evidence has also supported the oncologic advantage of LT. Dhir et al. [27] provided one of the earliest quantitative syntheses demonstrating significantly higher survival with LT compared to LR in early-stage HCC, thereby establishing a foundational rationale for transplant superiority in appropriately selected patients. More recently, López-López et al. [28] provided further insights, highlighting that patient selection and tumour biology remain determinative factors, and that LR may remain appropriate only for specific biological subgroups with intrinsically low recurrence risk. Together, these findings integrate the historical and contemporary evidence base, indicating that while the survival benefit of LT is firmly established within Milan-eligible early-stage cohorts, this advantage may also extend to selected intermediate-stage patients depending on tumour characteristics and recurrence risk. This broader context strengthens the present umbrella synthesis, demonstrating that LT provides the most durable oncologic control among eligible HCC patients.

These findings confirm LT as the preferred modality for transplant-eligible HCC patients within Milan criteria and highlight nuanced decision points. In patients with low microvascular invasion risk or HCV-related disease, LR may offer comparable outcomes in the short term, but long-term disease control is superior with LT [15,29]. Several studies have demonstrated that microvascular invasion significantly worsens post-LT recurrence risk and long-term survival, underscoring the importance of rigorous patient selection for resection versus transplantation [30,31,32]. Future research should focus on individual patient data meta-analysis integrating waitlist dynamics, donor source (living donor LT vs. deceased donor LT), and etiology.

The most critical clinical question raised by these findings is not whether LT confers superior oncologic outcomes compared with LR, but rather which patients should be prioritised for transplantation. Since the original Milan criteria proposed by Mazzaferro et al. [6], selection frameworks have progressively evolved from static morphologic thresholds toward more flexible, biology-oriented approaches. Contemporary studies have demonstrated that acceptable long-term outcomes can be maintained in selected patients beyond conventional Milan criteria, particularly when tumour biology, response to downstaging or bridging therapies, and dynamic disease behaviour are incorporated into decision-making [33,34].

While prior meta-analyses have consistently reported superior long-term outcomes with LT over LR, these comparisons have largely been framed at a procedural level. By integrating heterogeneous meta-analytic evidence within an umbrella framework, the present study enables a biology-informed reinterpretation of this established survival advantage. Specifically, the clinical relevance of transplantation superiority is not uniform across all patients with HCC. In patients with biologically indolent disease—characterised by favourable tumour biology, preserved liver function, or sustained response to locoregional therapies—LR may provide acceptable early outcomes and serve as a rational initial strategy.

In contrast, patients with biologically aggressive tumours are more likely to experience early recurrence following resection; in this context, LT offers a distinct oncologic advantage by addressing both tumour burden and the underlying cirrhotic substrate. Accordingly, the added value of this umbrella review lies in clarifying that the consistency of transplantation benefit across meta-analyses should be interpreted as a signal for biology-informed prioritisation, rather than as a mandate for universal transplantation. This reframing shifts the LT-versus-LR paradigm from a binary procedural comparison toward a clinically actionable, biology-adaptive strategy that aligns oncologic benefit with patient selection and organ allocation constraints.

Another important implication of the present findings is their potential impact on organ allocation and transplant system capacity. If LT is increasingly recommended on the basis of superior long-term oncologic outcomes, this could place additional strain on donor availability. However, this concern must be interpreted within the context of rapidly evolving transplantation practices. Recent evidence suggests that the introduction of machine perfusion technologies has significantly reduced waitlist times and expanded the effective donor pool, particularly for patients with lower Model for End-Stage Liver Disease (MELD) scores, such as those with HCC, who have traditionally experienced prolonged waiting periods [35].

At the same time, these gains may be counterbalanced by broader epidemiological trends. The global incidence of HCC is projected to rise substantially over the coming decades, driven by population ageing, metabolic liver disease, and ongoing viral hepatitis burden [36]. As a result, improvements in donor utilisation alone may not fully offset the increasing demand for transplantation. In this evolving landscape, our findings should not be interpreted as advocating indiscriminate expansion of transplantation eligibility. Rather, they support a prioritised and biology-informed application of LT, in which candidates most likely to derive durable oncologic benefit are preferentially allocated scarce grafts. Future allocation frameworks will need to integrate tumour biology, waitlist dynamics, donor preservation technology, and projected disease burden to ensure sustainable and equitable access to transplantation.

## 5. Conclusions

This PRIOR-compliant umbrella review (meta-analysis of meta-analyses) consolidates recent high-quality evidence and demonstrates that LT confers superior long-term overall and disease-free survival compared with LR in HCC. Through explicit overlap correction and separation of HR- and OR-based analyses, this study provides a reproducible and decision-grade evidence framework for surgical and transplant oncology. These findings support a selective, biology-informed application of transplantation that balances durable oncologic benefit against organ availability within evolving allocation systems.

## Figures and Tables

**Figure 1 cancers-18-00011-f001:**
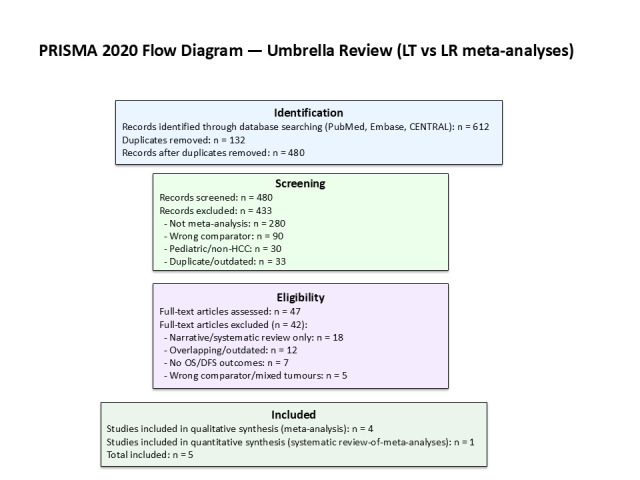
PRISMA 2020 flow diagram for the umbrella review.

**Figure 2 cancers-18-00011-f002:**
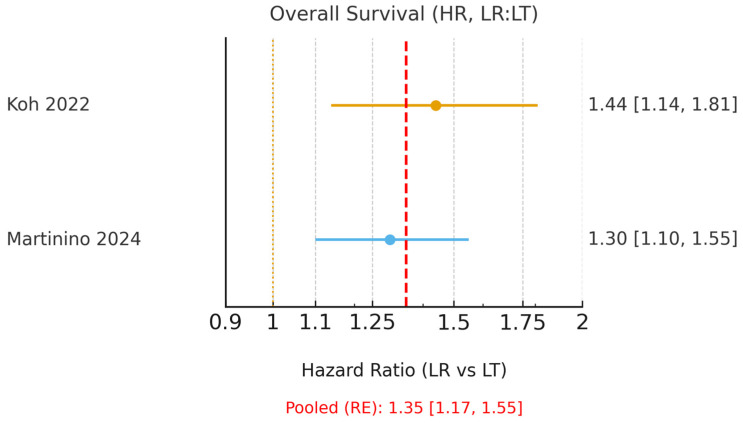
Forest plot of 5-year overall survival using hazard ratios (HR; LR = liver resection, LT = liver transplantation; values > 1 favour LT). A random-effects model was used to pool HRs from Koh (2022) [15] and the HR subset of Martinino (2024) [18]. The pooled HR with 95% CI, I^2^, τ^2^, and the corresponding 95% prediction interval are reported. The yellow dashed line indicates the line of no effect (HR = 1), and the red dashed line represents the pooled hazard ratio estimated using a random-effects model.

**Figure 3 cancers-18-00011-f003:**
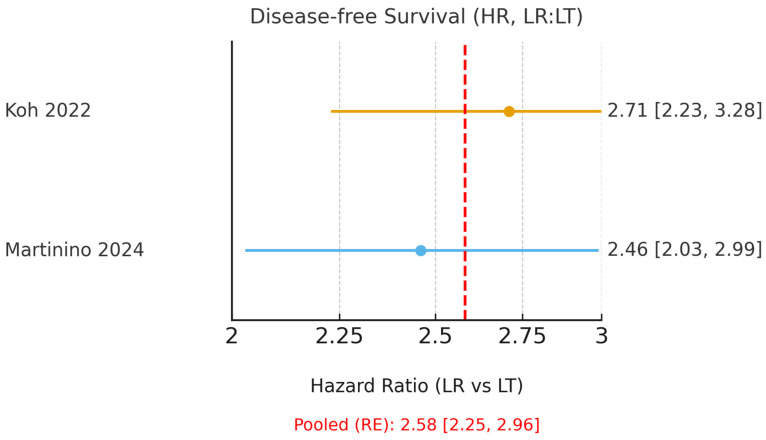
Forest plot of 5-year disease-free survival using hazard ratios (HR; LR = liver resection, LT = liver transplantation; values > 1 favour LT). A random-effects model was used to pool HRs from Koh (2022) [15] and the HR subset of Martinino (2024) [18]. The red dashed line indicates the pooled hazard ratio estimated using a random-effects model.

**Figure 4 cancers-18-00011-f004:**
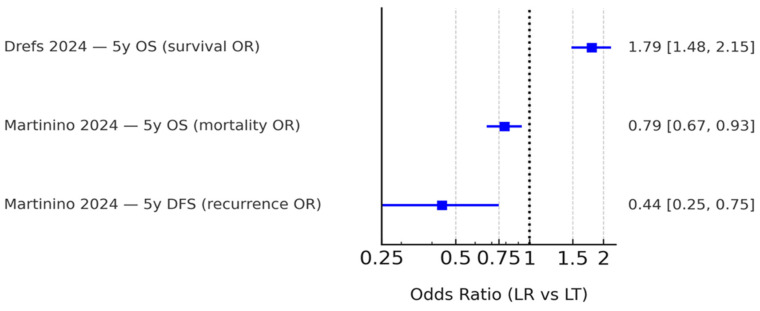
Summary of odds ratios (ORs) reported by eligible meta-analyses. Only Drefs (2024) [16] and Martinino (2024) [18] provided extractable quantitative OR data, while other reviews presented hazard ratio-based analyses or qualitative findings. Because event definitions varied (survival, mortality, and recurrence), these ORs are shown side-by-side without pooling. The black dashed line indicates the line of no effect (OR = 1).

**Table 1 cancers-18-00011-t001:** Summary of Published Meta-analyses Comparing Liver Transplantation and Liver Resection for hepatocellular carcinoma (2000–2025).

Author (Year)	Journal	Population Focus	Studies (*n*)	Patients (*n*)	Effect Measure	Key Findings	Ref. No.
Li HY (2012)	World J Gastroenterol	Salvage LT vs. LR	NR	NR	OR/HR	Salvage LT improved DFS	[10]
Xu XS (2014)	Hepatobiliary Pancreat Dis Int	All HCC	NR	NR	OR	LT > LR OS/DFS	[14]
Zheng Z (2014)	Transplantation	Observational studies	NR	NR	OR	LT > LR OS	[24]
Proneth A (2014)	Ann Surg Oncol	Cirrhotic HCC	NR	NR	HR	LT > LR	[25]
Schoenberg MB (2017)	Dtsch Arztebl Int	Early HCC	54	13,794 (LR 7990/LT 5804)	OR	LT > LR	[26]
Menahem B (2017)	Liver Transpl	Milan criteria/ITT	9	1431 (LR 570/LT 861)	HR	LT > LR (ITT)	[11]
Li W (2018)	Clin Transplant	HBV beyond Milan	NR	NR	OR	LT > LR	[12]
Kostakis ID (2019)	Transplant Proc	Salvage LT vs. repeat LR	NR	NR	OR	Salvage LT > repeat LR	[13]
Dhir M (2012)	HPB (Oxford)	Early HCC	NR	NR	OR/HR	LT > LR for OS/DFS	[27]
Koh JH (2022)	Hepatobiliary Surg Nutr	Milan criteria	35	18,421	HR	OS HR 1.44; DFS HR 2.71, LT > LR	[15]
Lopez-Lopez V (2024)	Langenbecks Arch Surg	BCLC B	NR	NR	OR	No conclusion for LT vs. LR (LT data too scarce)	[28]
Drefs M (2024)	Eur J Surg Oncol	All HCC, time-trend	63	19,804	OR	5y OS OR 1.79, LT > LR	[16]
Martinino A (2024)	Front Oncol	Meta-analysis of meta-analyses	10	—	OR/HR	LT > LR pooled	[18]
Rehman M (2025)	Cureus	SR-of-MAs (4 MAs)	4	>55,000 (across MAs)	Narrative	Confirmed consistency, AMSTAR-2 high	[19]
Bösch F (2025)	Asian J Surg	HBV/HCV focus	64	19,734	HR	LT > LR; transient 3y OS LR > LT in HCV	[17]

This table presents all 15 meta-analyses identified between 2000 and 2025, including classical and recent studies. Ten of these were already incorporated within the umbrella analysis conducted by Martinino et al. [18] and were therefore not duplicated in the quantitative pooling of the present study. Abbreviations: LT, liver transplantation; LR, liver resection; HCC, hepatocellular carcinoma; OS, overall survival; DFS, disease-free survival; HR, hazard ratio; OR, odds ratio; NR, not reported; HBV, hepatitis B virus; HCV, hepatitis C virus; ITT, intention-to-treat; AMSTAR-2, A Measurement Tool to Assess Systematic Reviews, version 2. Note: Reference numbers correspond to those in the main reference list.

**Table 2 cancers-18-00011-t002:** Key characteristics and findings of included meta-analyses.

Meta-Analysis	Type	Included Primary Studies	Total Patients	Primary Outcomes (5 y)	Key Effects	Ref. No.
Koh 2022 (HBSN)	MA	35	18,421	OS (HR), DFS (HR)	OS HR 1.44 (1.14–1.81); DFS HR 2.71 (2.23–3.28)	[15]
Drefs 2024 (EJSO)	MA	63	~19,804	OS (survival OR), RFS (OR)	OS OR 1.79 (1.48–2.15); RFS markedly higher after LT	[16]
Bösch 2025 (Asian J Surg)	MA	64	19,734	OS, RFS (3–5 y)	Overall LT > LR; HCV 3 y OS transient LR > LT (not at 5 y); RFS LT > LR in HCV	[17]
Martinino 2024 (Front Oncol)	Meta-meta	10 (MAs)	—	OS (OR), DFS (OR); HR subset	OS OR 0.79 (0.67–0.93); DFS OR 0.44 (0.25–0.75); OS HR 1.30; DFS HR 2.46	[18]
Rehman 2025 (Cureus)	SR-of-MAs	4 (MAs)	>55,000 (across MAs)	OS, DFS (narrative)	LT superior across MAs; AMSTAR-2 mostly high	[19]

Abbreviations: LT, liver transplantation; LR, liver resection; HCC, hepatocellular carcinoma; OS, overall survival; DFS, disease-free survival; HR, hazard ratio; OR, odds ratio; RFS, recurrence-free survival; MA, meta-analysis; SR-of-MAs, systematic review of meta-analyses; AMSTAR-2, A Measurement Tool to Assess Systematic Reviews, version 2. Note: Reference numbers correspond to those listed in the main reference section.

**Table 3 cancers-18-00011-t003:** Pooled hazard ratios (random-effects model) from the umbrella review (meta-analysis of meta-analyses), with heterogeneity statistics.

Outcome	k Meta-Analyses	Pooled HR [95% CI]	I^2^ (%)	τ^2^	95% Prediction Interval
Overall survival (HR, LR:LT)	2	1.35 [1.17–1.55]	0.0	0.00000	[1.11, 1.70]
Disease-free survival (HR, LR:LT)	2	2.58 [2.25–2.96]	0.0	0.00000	[2.00, 3.33]

Abbreviations: LT, liver transplantation; LR, liver resection; HR, hazard ratio; CI, confidence interval.

## Data Availability

The data that support the findings of this study are available upon reasonable request from the corresponding author.

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
