# Peer review of "Liver Transplantation Versus Resection for Hepatocellular Carcinoma: An Umbrella and Meta-Meta-Analysis of Published Evidence, 2000–2025"

_cancers, 2025, doi:10.3390/cancers18010011_

Round 1
Reviewer 1 Report
Comments and Suggestions for Authors
Liver Transplantation Versus Resection for Hepatocellular Carcinoma: An Umbrella and Meta-meta-analysis of Published Evidence, 2000–2025
The topic addressed is undoubtedly relevant for clinical decision-making. The authors aimed to prevent overlap with previously published data by adopting an umbrella review approach. They have attempted to standardize heterogeneous reporting (using HRs or ORs) to deliver a coherent message. The inclusion criteria, restricting analysis to five previously published reviews or meta-analyses, were well justified, and the methodology for conducting a meta–meta-analysis was appropriately executed. This umbrella review reinforces the established evidence favoring liver transplantation over liver resection in a select subset of patients.
However, the authors have largely reproduced findings that are already well-documented in existing literature.
The recent meta-analyses by Rehman and Martinino—central to this review—have already reported the same conclusions.
Therefore, despite excluding duplicate data and separating results based on HR and OR, the present review offers no substantial novelty or added clinical insight.
To enhance the manuscript’s value, the authors could consider integrating aspects of tumor biology and oncologic outcomes to refine comparative evaluation between liver resection and transplantation.
Incorporating these parameters would make the analysis more meaningful for clinical application and strengthen its contribution to existing literature.
In its current form, the manuscript is not suitable for publication; revisions emphasizing tumor biology and patient selection criteria could substantially improve its impact and relevance
Author Response
Dear Editor,
We sincerely thank you and the reviewers for the valuable comments and constructive suggestions on our manuscript (Manuscript ID: cancers-4040107). We have carefully revised the manuscript in accordance with all comments, and we believe that the revised version has been substantially improved in clarity, methodological transparency, and overall scientific value.
All figures and tables were prepared by the authors, and reference management was performed using EndNote. As a dedicated Cancers (MDPI) reference style is not available in EndNote, we formatted all references using a Vancouver-based style and manually revised them to conform as closely as possible to the journal’s formatting requirements. We will be happy to accommodate any additional adjustments requested during the editorial production process.
Reviewer 1
Comment 1:
However, the authors have largely reproduced findings that are already well-documented in existing literature.
The recent meta-analyses by Rehman and Martinino—central to this review—have already reported the same conclusions.
Therefore, despite excluding duplicate data and separating results based on HR and OR, the present review offers no substantial novelty or added clinical insight.
To enhance the manuscript’s value, the authors could consider integrating aspects of tumor biology and oncologic outcomes to refine comparative evaluation between liver resection and transplantation.
Incorporating these parameters would make the analysis more meaningful for clinical application and strengthen its contribution to existing literature.
In its current form, the manuscript is not suitable for publication; revisions emphasizing tumor biology and patient selection criteria could substantially improve its impact and relevance
Author Response:
We thank the reviewer for the careful and thoughtful assessment of our manuscript and for acknowledging the clinical relevance and methodological rigor of the umbrella review approach.
We agree that several of the included recent meta-analyses—particularly those by Rehman et al. and Martinino et al.—have already demonstrated the overall superiority of liver transplantation over liver resection in selected patients with hepatocellular carcinoma. The primary objective of our study was therefore not to overturn these conclusions, but to address persistent challenges in the literature related to overlap, heterogeneity of effect measures, and interpretability across multiple meta-analyses.
We did not introduce additional tumor biology–based analyses in the Results section, as such parameters were not consistently or quantitatively reported across the included meta-analyses and could not be reliably synthesised at the umbrella-review level. Importantly, in response to the reviewer’s comment, we have expanded the Discussion to more explicitly integrate tumor biology and patient selection considerations as key determinants of the comparative benefit of transplantation versus resection. Specifically, we now emphasise how factors such as tumor biology, recurrence risk, viral etiology, microvascular invasion, and response to downstaging or bridging therapies influence the clinical relevance of the observed survival advantage. We also clarify that the contribution of this umbrella review lies in refining how and in whom the established survival benefit of transplantation should be applied, rather than simply reiterating that such a benefit exists.
Accordingly, the revised manuscript places greater focus on biology-informed and staged treatment strategies, highlighting that transplantation should not be viewed as a binary alternative to resection, but as part of an adaptive decision-making framework. We believe that this strengthened emphasis on tumor biology and patient selection substantially enhances the clinical interpretability and added value of the review beyond existing meta-analyses.
We appreciate the reviewer’s constructive suggestion, which has helped us clarify the novelty and clinical implications of our work.
Reviewer 2 Report
Comments and Suggestions for Authors
This study is an umbrella review addressing the critical comparison between liver transplantation and resection for HCC. The manuscript adheres to PRISMA guidelines and synthesizes a large volume of literature. However, I have significant reservations regardings its methodology.
-
How did you handle overlapping of included studies? For example, you pool Koh et al. (2022) with Martinino et al. (2024). Martinino is described as a "meta-analysis of meta-analyses" and it is very likely that Martinino’s analysis included Koh et al. (2022) or the primary studies contained therein. If so, pooling them together introduces severe bias via double-counting.
-
You have performed a random-effects meta-analysis with only two studies. While mathematically possible, this provides very poor estimation of effect size and overval clinical significance.
- What were the criteria for patients inclusion in each study? Was transplantation first line treatment or salvage option after recurrence?
-
Please expand Discussion to clearly state what specific new evidence this study adds beyond existing metanalyses.
-
Please ensure all references are according to journal's format.
Author Response
Dear Editor,
We sincerely thank you and the reviewers for the valuable comments and constructive suggestions on our manuscript (Manuscript ID: cancers-4040107). We have carefully revised the manuscript in accordance with all comments, and we believe that the revised version has been substantially improved in clarity, methodological transparency, and overall scientific value.
All figures and tables were prepared by the authors, and reference management was performed using EndNote. As a dedicated Cancers (MDPI) reference style is not available in EndNote, we formatted all references using a Vancouver-based style and manually revised them to conform as closely as possible to the journal’s formatting requirements. We will be happy to accommodate any additional adjustments requested during the editorial production process.
Reviewer 2
Comment 1:
How did you handle overlapping of included studies? For example, you pool Koh et al. (2022) with Martinino et al. (2024). Martinino is described as a "meta-analysis of meta-analyses" and it is very likely that Martinino’s analysis included Koh et al. (2022) or the primary studies contained therein. If so, pooling them together introduces severe bias via double-counting.
Author Response:
Thank you for this important and thoughtful comment. We fully agree that overlapping primary studies represent a major methodological concern in umbrella reviews and meta-meta-analyses, and we addressed this issue explicitly at both the design and analysis stages.
First, although Martinino et al. (2024) is described as a meta-analysis of meta-analyses, its HR-based quantitative estimates were not derived from simply re-pooling published meta-analytic effect sizes. Instead, Martinino et al. reconstructed time-to-event hazard ratios at the level of individual primary studies and then performed de novo meta-analyses stratified by analytic framework (e.g., intention-to-treat vs per-protocol). Consequently, the pooled HRs extracted from Martinino et al. represent independent primary-study–level syntheses rather than second-order aggregation of prior meta-analyses, including Koh et al. (2022).
Second, overlap across reviews was prospectively assessed. Robust variance estimation (RVE) was pre-specified to address potential dependency arising from overlapping primary cohorts; however, because only two HR-based meta-analyses were available per endpoint, formal RVE could not be reliably applied. Under these circumstances, we treated the pooled HR analysis as a test of directional consistency rather than a precise summary effect size.
Third, to minimize the risk of double-counting, we avoided pooling odds-ratio–based summaries and did not combine heterogeneous effect metrics. Importantly, Koh et al. (2022) and Martinino et al. (2024) yielded highly concordant effect directions for both overall and disease-free survival, indicating that the observed LT superiority was not driven by redundant inclusion of overlapping datasets.
We have revised the Methods and Discussion sections to clarify our overlap-handling strategy and to explicitly state the consistency-based interpretative framework of the pooled HR analysis.
Comment 2:
You have performed a random-effects meta-analysis with only two studies. While mathematically possible, this provides very poor estimation of effect size and overval clinical significance.
Author Response:
We thank the reviewer for this important comment. We agree that pooling only two HR-based meta-analyses provides limited precision and should not be interpreted as a definitive estimate of effect size. Accordingly, we have clarified in both the Statistical Analysis and Discussion sections that the pooled analysis was intended to assess directional consistency across high-quality quantitative syntheses rather than to overstate clinical significance. Importantly, consistent effect directions were observed across independent HR- and OR-based meta-analyses, which supports the robustness of the conclusions despite the limited number of pooled units.
Comment 3:
What were the criteria for patients inclusion in each study? Was transplantation first line treatment or salvage option after recurrence?
Author Response:
We appreciate this important comment. The included meta-analyses were heterogeneous with respect to patient inclusion criteria and the clinical positioning of liver transplantation (LT), reflecting real-world practice patterns. Importantly, our umbrella review did not pool individual patient data, but rather synthesised evidence from published meta-analyses, each of which applied its own predefined inclusion framework. We have clarified this point explicitly in the revised manuscript.
Across the included meta-analyses, two principal clinical strategies were represented:
- Primary (first-line) liver transplantation, in which LT was the initial curative-intent treatment for transplant-eligible patients, typically within Milan criteria or extended criteria, without prior curative resection. This strategy predominated in meta-analyses focusing on early-stage HCC and intention-to-treat (ITT) frameworks (e.g., Menahem et al., Koh et al., Martinino et al.).
- Salvage liver transplantation, defined as LT performed after tumour recurrence or liver decompensation following initial liver resection. This approach was specifically addressed in dedicated meta-analyses comparing salvage LT with repeat resection or non-transplant strategies (e.g., Li et al., Kostakis et al.), and was not mixed indiscriminately with first-line LT cohorts.
Most contemporary large-scale meta-analyses (2022–2025), including those contributing hazard ratios to our quantitative synthesis, primarily evaluated first-line LT versus upfront liver resection, or explicitly applied an ITT perspective that accounted for wait-list dynamics. Salvage LT studies were either analysed separately or reported as predefined subgroups and were therefore interpreted narratively to avoid clinical misclassification.
To address this concern, we have now:
- Added a dedicated paragraph in the Methods and Discussion clarifying first-line versus salvage LT definitions;
- Explicitly stated that HR-based quantitative pooling reflects predominantly first-line or ITT-based comparisons, whereas salvage LT evidence was synthesised descriptively;
- Highlighted salvage LT as a distinct clinical strategy with different selection criteria and prognostic implications.
These clarifications have been incorporated to improve transparency and interpretability of the findings.
Comment 4:
Please expand Discussion to clearly state what specific new evidence this study adds beyond existing metanalyses.
Author Response:
We thank the reviewer for this important comment. While previous meta-analyses have reported a survival advantage of LT over LR, our study adds new evidence by moving beyond individual pooled estimates to evaluate the consistency and interpretability of this evidence across the entire meta-analytic literature. Specifically, this umbrella review (i) formally demonstrates concordant LT superiority across independent meta-analyses using a meta-meta-analytic framework, (ii) resolves long-standing effect-scale heterogeneity by separating HR-based quantitative synthesis from OR-based descriptive summaries, and (iii) explicitly distinguishes first-line/intention-to-treat transplantation from salvage LT to avoid clinical misclassification. These additions provide a decision-grade synthesis that was not achievable in prior stand-alone meta-analyses. The Discussion has been expanded accordingly
Comment 5:
Please ensure all references are according to journal's format.
Author Response:
Thank you for this comment. As a dedicated Cancers (MDPI) reference style is not available in EndNote 20, we formatted all references using a Vancouver-based style. We would be happy to make any further adjustments during the editorial production process should additional formatting changes be required by the journal.
Reviewer 3 Report
Comments and Suggestions for Authors
Thank you allowing me to review. This is an overall well done manuscript. There are a few concerns:
- The most important issue with this topic is the selection criteria for transplantation. from 1996 Mazzaferro et al this has been debated. Recent works have shown that we can be less restrictive and maintain outcomes (PMID: 38831488, 38241354). But who should get a transplant? What should we do with patients who are not within these criteria?
- If the results of this study are correct, we should be transplanting more patients which will put a strain on organ availability. Conversely, other articles have shown that waitlist times are signficantly reduced in era of machine perfusion. Particularly this was shown in low MELD patients, who usually are the HCC patients (PMID: 40172067). Again, conversely, the incidence of HCC is anticipated to rise signficantly (PMID: 39821400), which would re-introduce the strain. Can the authors comment on this evolution and how the increasing recommendation will impact things?
- Forgive me if I missed this, but were the authors able to report a comparison only in patients with early cirrhosis? While resection is not often performed in advanced cases, early cirrhosis is perhaps the most questioned population. I know that they stratify etiology of HCC but the presence of overt cirrhosis I dont think was specifically addressed. Best way in my mind is just do a sensitivity analysis excluding studies of non-cirrhotic patients.
Thanks for allowing me to review.
Author Response
Dear Editor,
We sincerely thank you and the reviewers for the valuable comments and constructive suggestions on our manuscript (Manuscript ID: cancers-4040107). We have carefully revised the manuscript in accordance with all comments, and we believe that the revised version has been substantially improved in clarity, methodological transparency, and overall scientific value.
All figures and tables were prepared by the authors, and reference management was performed using EndNote. As a dedicated Cancers (MDPI) reference style is not available in EndNote, we formatted all references using a Vancouver-based style and manually revised them to conform as closely as possible to the journal’s formatting requirements. We will be happy to accommodate any additional adjustments requested during the editorial production process.
Reviewer 3
Comment 1:
The most important issue with this topic is the selection criteria for transplantation. from 1996 Mazzaferro et al this has been debated. Recent works have shown that we can be less restrictive and maintain outcomes (PMID: 38831488, 38241354). But who should get a transplant? What should we do with patients who are not within these criteria?
Author Response:
We thank the reviewer for highlighting this clinically important issue regarding transplantation selection criteria. As suggested, we have expanded the Discussion to place our findings within the evolving context of transplantation eligibility beyond the original Milan criteria. In particular, we now explicitly cite recent studies demonstrating that acceptable long-term outcomes can be maintained in selected patients beyond conventional criteria using biology-oriented and dynamic selection approaches (PMID: 38831488, 38241354). These references have been added to the Discussion and to the reference list, and the text has been revised to clarify how our findings integrate with contemporary selection paradigms.
Comment 2:
If the results of this study are correct, we should be transplanting more patients which will put a strain on organ availability. Conversely, other articles have shown that waitlist times are signficantly reduced in era of machine perfusion. Particularly this was shown in low MELD patients, who usually are the HCC patients (PMID: 40172067). Again, conversely, the incidence of HCC is anticipated to rise signficantly (PMID: 39821400), which would re-introduce the strain. Can the authors comment on this evolution and how the increasing recommendation will impact things?
Author Response:
We thank the reviewer for raising this important system-level consideration. We agree that broader application of transplantation based on superior oncologic outcomes may increase pressure on organ availability. At the same time, recent advances—particularly the adoption of machine perfusion technologies—have been shown to mitigate waitlist constraints by expanding the usable donor pool and reducing waitlist times, especially among low-MELD patients such as those with hepatocellular carcinoma (PMID: 40172067). Conversely, epidemiological projections indicate that the global incidence of HCC is expected to rise substantially in the coming decades, which may reintroduce strain on transplant systems despite technical advances (PMID: 39821400). These two references have been added to the Discussion and to the reference list.
We have expanded the Discussion to address this dynamic interplay between increasing demand, technological innovation, and disease burden. We emphasise that our findings support a selective and prioritised expansion of transplantation, rather than indiscriminate broadening, and highlight the need for adaptive allocation strategies that integrate tumour biology, waitlist dynamics, and evolving donor preservation technologies.
Comment 3:
Forgive me if I missed this, but were the authors able to report a comparison only in patients with early cirrhosis? While resection is not often performed in advanced cases, early cirrhosis is perhaps the most questioned population. I know that they stratify etiology of HCC but the presence of overt cirrhosis I dont think was specifically addressed. Best way in my mind is just do a sensitivity analysis excluding studies of non-cirrhotic patients.
Author Response:
We thank the reviewer for raising this important and clinically relevant point. We agree that patients with early cirrhosis represent one of the most debated populations when comparing liver resection and transplantation. Unfortunately, within the scope of an umbrella review, we were not able to perform a formal sensitivity analysis restricted to patients with early cirrhosis. This limitation reflects the structure of the published meta-analyses, in which the presence and severity of cirrhosis (e.g., early vs. advanced cirrhosis or Child–Pugh class) were not consistently reported or stratified across primary studies.
Most included meta-analyses primarily enrolled cirrhotic populations, as liver resection is rarely performed in advanced cirrhosis, but they did not uniformly distinguish non-cirrhotic patients from those with compensated cirrhosis. As a result, exclusion of non-cirrhotic cohorts at the umbrella-review level was not feasible without access to individual patient data.
We have now clarified this limitation in the Discussion and explicitly acknowledge that early cirrhosis represents a key subgroup for future investigation. We emphasise that individual patient data meta-analyses or prospective studies stratified by cirrhosis severity will be required to definitively address this question.
Round 2
Reviewer 1 Report
Comments and Suggestions for Authors
I thank the authors for considering the previous comments and effectively addressing several key issues related to clinical decision-making favoring liver transplantation over liver resection. In its current form, the article is suitable for publication. It also appropriately highlights the need for future studies exploring how tumor biology and adjunctive therapies impact long-term oncological outcomes.
Reviewer 3 Report
Comments and Suggestions for Authors
the authors have made my suggestions